# TABBY: TABULAR DATA SYNTHESIS WITH LANGUAGE MODELS

## ABSTRACT

While advances in large language models (LLMs) have greatly improved the quality of synthetic text data in recent years, synthesizing *tabular* data has received far less attention. Many of the top-performing approaches to this problem rely on techniques that adapt models originally developed for other modalities, potentially leaving generative performance on the table. We address these disparities in attention and performance for tabular data by introducing **Tabby**, a simple but powerful post-training modification to the standard Transformer-based language model architecture that enables its use for tabular dataset synthesis. Tabby relies on Gated Mixture-of-Experts layers, allowing each data column to be modeled by a dedicated set of parameters within the transformer multi-layer perceptrons or language modeling heads. Applying Tabby to Distilled-GPT2 improves synthetic data quality up to 7% compared to previous tabular dataset synthesis methods, achieving performance near or equal to that of real data.

## 1 INTRODUCTION

From spreadsheets to databases, much of our modern life is encoded in tables. Airplane black boxes, website visitor logs and hospital patient records are just a few examples of this versatile modality. Despite the widespread use of tabular data and many calls for improved tabular modeling approaches, this type of data has received less attention in deep learning research than images and text (Fang et al., 2024; Davila et al., 2024; van Breugel and van der Schaar, 2024).

Progress towards the synthesis of realistic tabular data has encountered several key challenges. First, tabular columns often exhibit complex interdependencies. Second, many tabular datasets are in fact a combination of various modalities, with text, numerical, and nested datatypes (such as a JSON, dictionary, or other structured object) possible among the columns in one dataset. Third, although the order of items within one column of one row is important, the order of columns with respect to each other is usually not meaningful and is a potential source of spurious correlations when training a model. How to best design and train models that can address these issues remains an open question.

There have been notable efforts to adapt several model architectures to tabular data, recently focusing on generative adversarial networks (GANs) (Xu et al., 2019), LLMs (Borisov et al., 2022) and diffusion models (Kotelnikov et al., 2022). However, because these architectures were each designed with text or images in mind, significant preprocessing must be made to tabular datasets in order to allow their usage, likely resulting in lower performance than would be possible for an architecture designed specifically for tabular data.

For these reasons, works including van Breugel and van der Schaar (2024) have called for the development of pretrained *Large Tabular Models (LTMs)* to fill a similar role to text and image foundation models, such as GPT (OpenAI, 2023) or DALL-E (OpenAI, 2021). Unfortunately, the creation of an LTM would require (1) large and diverse tabular pretraining sets which have not yet been curated, (2) a specialized tabular model architecture which has yet to be designed, along with (3) a staggering amount of compute resources for pretraining.

In this work, we take an initial step towards the development of a LTM by proposing ***Tabby**, a post-training modification to the transformer-based LLM architecture for enabling tabular data synthesis*. After training on text data—but before finetuning on tabular data—Tabby replaces designated LLM blocks with *Mixture-of-Experts (MoE) layers* (Shazeer et al., 2017), which allow each data column to be modeled by a dedicated set of parameters within the LLM. The greater model expressivity afforded by this modification results in higher-fidelity synthetic data.

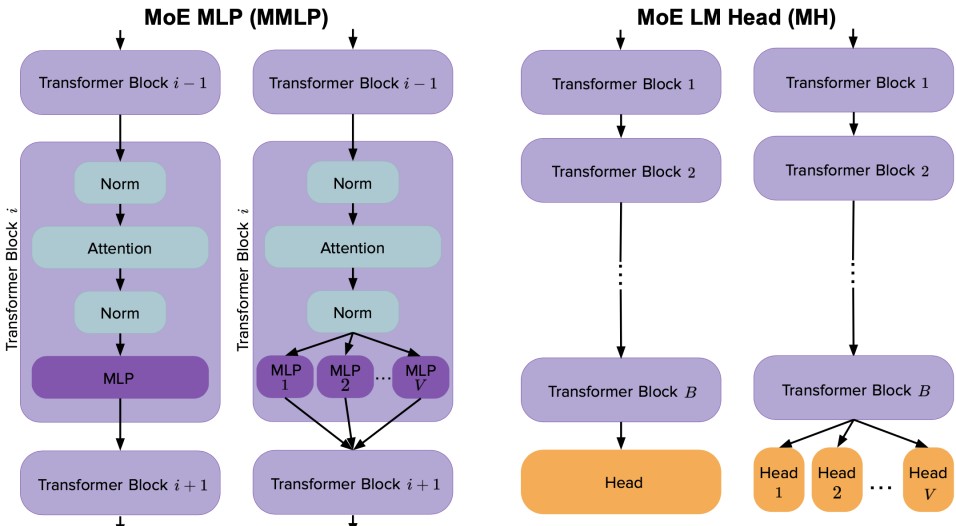

Figure 1: An overview of the Tabby MMLP (left) and MH (right) modifications.

To our knowledge, Tabby is *the first architecture modification to make LLMs better-suited to table generation*. Using a pretrained LLM as a starting point allows Tabby to take advantage of its diverse text pretraining, avoiding the logistical challenges of training a LTM entirely from scratch. We find that, according to standard metrics, **Tabby produces synthetic data near- or at-parity with real tabular data on 3 out of 6 datasets**, a level of performance not achieved by prior works. We summarize our contributions as follows:

- We introduce Tabby, the first architecture modification that allows transformer-based LLMs to synthesize more realistic tabular data.
- We explore multiple tabular training techniques for LLMs, including our *Plain training method*: a simple, lightweight training technique that may serve as an effective baseline for training future LLM-based tabular synthesis approaches.
- We demonstrate that Tabby produces higher-quality synthetic data for 4 out of 6 datasets, and also allows greater insights into the model's performance and training progress than other tabular synthesis approaches.

After an exploration of prior tabular synthesis approaches in Section 2, we provide more details on Tabby in Section 3. Next, we conduct extensive experiments in Section 4.

## 2 RELATED WORK

Tabular data has played a central role in machine learning since the field's early days. In particular, decision trees (Song and Lu, 2015) and their relatives such as random forests (Biau and Scornet, 2016) are well-adapted to classification or regression on tabular datasets.

**Classical synthesis:** Classical machine learning methods may be used to synthesize tabular data, by modeling each column as a random variable and sampling from the resulting multivariate distribution. This technique has been successfully applied to decision trees (Reiter, 2005) and Bayesian networks (Aviñó et al., 2018; Zhang et al., 2017) . Copulas (Frees and Valdez, 1998; Janke et al., 2021; Benali et al., 2021) are another traditional approach, which rely on first modeling each column as a univariate random distribution, then fitting a probabilistic model to the multivariate distribution formed by all columns. However, these approaches are limited in the data types that may be represented among the columns and the varieties of relationships that may be modeled across columns.

**Generative Adversarial Networks (GANs):** Many tabular synthesis methods rely on GANs (Goodfellow et al., 2020), but have encountered several inherent limitations. In particular, the

distributions of ordinal columns are frequently rather imbalanced, leading GANs to undesirable phenomena such as mode collapse. Continuous columns may possess multiple modes and complex interactions with the other columns which GANs also struggle to capture. The top-performing CT-GAN (Xu et al., 2019) employs conditional generation in an effort to address these shortcomings. However, the fidelity of CTGAN's synthetic data leaves further improvements to be desired, as we will demonstrate in Section 4.

**LLMs:** A small body of work has sought to apply LLMs' demonstrated abilities of modeling complex relationships to tabular data. The landmark work in this area, GReaT (Borisov et al., 2022), details methods to convert tabular data into a sentence format which may be input to LLMs, then proposes a training technique of "shuffling" the order in which columns occur for each row, which is reported to improve the modeling of inter-column dependencies.

Two notable works have built off of GReaT to achieve further improved tabular data fidelity: TapTap (Zhang et al., 2023) pretrains full or Distilled-GPT2 (Radford et al., 2019) on a variety of tabular datasets before fine-tuning on a downstream tabular synthesis task, while Tabula (Zhao et al., 2023) explores methods of preprocessing the training data to be more-easily modeled by LLMs. Other LLM-based works have adapted these recent advances to relational tables (Solatorio and Dupriez, 2023), or used the emergent abilities of very large models such as GPT-4 to generate synthetic data using In-Context Learning in place of fine-tuning (Seedat et al., 2024). Because many of these prior LLM-based works are training techniques, they may be applied in concert with the Tabby architecture modification. We demonstrate this using GReaT, TapTap and Tabula in Section 4.

**MoE Architectures:** The key innovation of Tabby is the application of Gated Mixture of Expert (MoE) layers (Shazeer et al., 2017; Masoudnia and Ebrahimpour, 2014) for LLM synthesis of tabular data. MoE layers have enjoyed utility in multitask (Ma et al., 2018; Gupta et al., 2022) and multimodal learning (Zhao et al., 2024; Park et al., 2018), by creating sets of model parameters that are dedicated to a specific task. We describe our use of MoE layers in Section 3.

## 3 METHOD

Tabby is an architecture modification that may be applied to any transformer-based language model (LM) (Vaswani, 2017). In Section 3.1, we describe the variations of Tabby. In Section 3.2 , we describe the process for training an arbitrary LM on tabular data, then compare the training process's forward pass and loss calculation for a Tabby model with a non-Tabby model in Section 3.3. Tabby increases the expressivity of LMs, allowing for better modeling of individual columns' distributions and resulting in higher generative fidelity.

### 3.1 ARCHITECTURE OF TABBY MODELS

Suppose that a tabular dataset contains $V$ columns and let the order of blocks within an arbitrary transformer-based LM be represented as $[L_1, L_2, \ldots, L_H]$. We apply the MoE technique by replacing an LM block $L_a$ with a vector $\Lambda_a = [L_{a,1}, L_{a,2}, \ldots, L_{a,V}]$ of $V$ blocks. As such, a Tabby model with one MoE block $\Lambda_a$ is represented as

$$[L_1, L_2, \ldots, L_{a-1}, [L_{a,1}, L_{a,2}, \ldots, L_{a,V}], L_{a+1}, \ldots, L_H].$$

The $i$-th column in the dataset is modeled by $L_{a,i}$ within $\Lambda_a$.

Tabular data comes in myriad forms, such as relational, multidimensional or sparse tables. The flexibility in placement of MoE layers enables Tabby's adaptation to a variety of table setups. While this technique may be applied to any set of layers within the model, we focus on the transformer blocks' multi-layer perceptrons (MLPs) and the language modeling (LM) head. We refer to Tabby models with MoE MLPs or LM heads as MMLP or MH models, respectively. Those with both MoE MLPs and MoE LM Heads are MMLP-MH models. For a visual comparison of MMLP, MH and non-Tabby models, refer to Figure 1.

### 3.2 FINE-TUNING LLMS ON TABULAR DATA

We now describe the general conventions applied when training or fine-tuning an LM on a tabular dataset. Suppose our training dataset contains $N$ rows and the dataset's column names are denoted

by $v_1, v_2, \ldots, v_V$, where the value of the $j$-th row in the $i$-th column is denoted as $v_i^j$. For a given row, the model will train on the columns in order $\ell_1, \ell_2, \ldots, \ell_V$ (for general training we consider this order to be simply $[V]$, while GReaT training allows this order to be arbitrary).

To provide the LM with its expected text modality input, we convert the $j$-th row as follows, where `<EOS>` is the end-of-sequence token and `<EOC>` is a specialized end-of-column token which we introduce to divide the text between columns:

$$``\texttt{<BOS>}\ v_{\ell_1}\ \texttt{is}\ v_{\ell_1}^j\ \texttt{<EOC>}\ v_{\ell_2}\ \texttt{is}\ v_{\ell_2}^j\ \texttt{<EOC>}\ \cdots\ v_{\ell_V}\ \texttt{is}\ v_{\ell_V}^j\ \texttt{<EOS>}"$$

After converting the tabular dataset in this fashion, an LM is capable of fine-tuning on the dataset in a normal sequence-to-sequence style. The prompt for each row during training is the beginning-of-sequence token `<BOS>`. During generation, the LM will output text in a similar format to the training data, which can then be parsed into tabular data as desired.

## 3.3 Tabby Training

Suppose that we construct a Tabby model from a base LM by replacing one of its blocks $L_a$ with an MoE set $\Lambda_a$. At the beginning of fine-tuning the Tabby model, the weights for each block in $\Lambda_a$ are initialized to equal the weights of $L_a$.

The Tabby training process requires only slight modifications as compared to the training of other LMs on tabular data. Instead of representing each training row as one string, we convert each row into a length-$V$ list of strings as follows:

$$[``v_{\ell_1}\ \texttt{is}\ v_{\ell_1}^j\ \texttt{<EOC>}",\ ``v_{\ell_2}\ \texttt{is}\ v_{\ell_2}^j\ \texttt{<EOC>}",\ \cdots,\ ``v_{\ell_V}\ \texttt{is}\ v_{\ell_V}^j\ \texttt{<EOS>}"]$$

Internally, the Tabby model begins by training on column $\ell_1$ with prompt `<BOS>`, attending to tokens 0 through $k-1$ when predicting the $k$-th token. After computing the loss on column $\ell_1$, this column's tokens are appended to the prompt used to train column $\ell_2$. As such, the prompt when training on column $\ell_i$ is

$$``\texttt{<BOS>}v_{\ell_1}\ \texttt{is}\ v_{\ell_1}^j\ \texttt{<EOC>}\ v_{\ell_2}\ \texttt{is}\ v_{\ell_2}^j\ \texttt{<EOC>}\ \cdots\ v_{\ell_{i-1}}\ \texttt{is}\ v_{\ell_{i-1}}^j\ \texttt{<EOS>}"$$

A favorable side-effect of this training style is that we calculate the losses for each column separately, allowing the performances of each column to be monitored separately and compared, as demonstrated in Section 4.3.

# 4 Experimental Results

With our evaluations, we seek to assess the following claims:

**Claim 1**: Tabby models generate higher-quality data than pre-existing tabular synthesis approaches.
**Claim 2**: The Tabby architecture modification allows smaller LLMs to achieve synthetic data fidelity more similar to that of LLMs with higher parameter counts.
**Claim 3**: Tabby's loss formulation allows for convenient tracking of per-column performance at training time, leading to better understanding of model behavior.

After providing evaluation setup details in Section 4.0, we compare Tabby to a broad array of prior works on diverse tabular datasets in Section 4.1 to evaluate Claim 1. As Tabby may be applied to any transformer-based LM, we explore Claim 2 for LMs of highly disparate sizes in Section 4.2. Lastly, in Section 4.3, we investigate how Tabby adapts to individual columns within a dataset during finetuning as a demonstration of Claim 3.

Table 1: Summary statistics of datasets. The first three columns list the number of rows in each data split, while the next two columns display the number of categorical versus numerical features, respectively. The rightmost column details whether the dataset is considered a classification (C) or regression (R) task in downstream evaluations.

|  | N Train | N Validation | N Test | # Cat. | # Num. | Task |
|---|---|---|---|---|---|---|
| Diabetes (Kahn, 1994) | 576 | 57 | 135 | 0 | 8 | C |
| Travel (Tejashvi, 2023) | 715 | 71 | 168 | 4 | 2 | C |
| Adult (Becker and Kohavi, 1996) | 36631 | 3663 | 8548 | 8 | 6 | C |
| Abalone (Nash et al., 1994) | 3132 | 313 | 732 | 1 | 7 | R |
| Rainfall (Zaman, 2018) | 12566 | 1256 | 2933 | 2 | 1 | R |
| House (Pace and Barry, 1997) | 15480 | 1548 | 3612 | 0 | 8 | R |

## 4.0 SETUP

We now detail our experiments' baselines, evaluation datasets and metrics.

### 4.0.1 BASELINES AND COMPARISONS

**LLM Approaches:** As previously described, there are multiple approaches to training LLMs on tabular data, regardless of whether Tabby is applied. As a baseline training technique, we implement *Plain* training. While this method has not been described in prior LLM works, it represents a basic method of training the LLM on the columns in the same order as they are found in the training dataset. At sample time, we simply prompt with <BOS> and parse the resulting model output.

Next, we explore the GReaT technique (Borisov et al., 2022) as introduced in Section 2. At each step, the order in which the columns are presented to the model are selected at random. During generation, GReaT enforces that the distribution of the label column matches that of the training distribution. Suppose that the label column is $v_t$ and let $S_t$ be the set of all unique values taken by $v_t$ in the trainset (regardless of whether $v_t$ is a categorical or numerical column). GReaT prompts the LLM with <BOS>$v_t$ is $s_t$, where $s_t$ is sampled from $S_t$ with probability proportionate to the frequency of $s_t$ in column $v_t$.

We additionally explore the use of two more training techniques in conjunction with GReaT. TapTap (Zhang et al., 2023) is a checkpoint of Distilled-GPT2, pretrained using GReaT on a large collection of tabular datasets. Meanwhile, Tabula (Zhao et al., 2023) aims to address the challenges encountered by LLMs on categorical columns: Tabula converts each categorical column into an ordinal format by replacing each unique value of the column with a unique integer. In many cases, this technique reduces sequence length, decreasing training and generation time, and helps the LLM during sampling to only generate values that occur within the categorical column's training distribution. We abbreviate the training system of using GReaT, TapTap and Tabula together as GTT.

All LLM methods use Distilled-GPT2 as a base model, save for the GTT methods which use the TapTap Distilled-GPT2 that is pretrained on tabular data as a base model. We finetune Non-Tabby (NT) Distilled-GPT2 using Plain training as a baseline.

**Other Approaches:** To represent non-LLM tabular synthesis techniques, we include a diffusion model, Tab-DDPM (Kotelnikov et al., 2022), as well as CTGAN (Xu et al., 2019) and TVAE (Xu et al., 2019), the leading GAN and VAE approaches, respectively.

### 4.0.2 DATASETS

We evaluate Tabby on six common tabular datasets, which are summarized in Table 1. The majority of these datasets are standard for the evaluation of tabular synthesis techniques, allowing for easy comparison with prior approaches. For more information on these datasets, see Appendix A.

### 4.0.3 METRICS

We focus on the standard metrics of machine learning efficacy and detection accuracy to measure the fidelity and quality of synthetic datasets.

*Machine learning efficacy (MLE)* (Dankar et al., 2022) quantifies whether a synthetic dataset is capable of replacing the original, real data used to train a generative model. MLE serves as our primary metric for data quality. Given a real dataset, we form disjoint training and test sets, denoted $R$ and $D$ respectively. A generative model is trained on $R$, then generates synthetic dataset $S$.

To calculate MLE, a downstream classifier or regressor $K_R$ is trained using $R$ to predict a predetermined label column, using all other columns as features. An additional classifier or regressor $K_S$ is similarly trained on $S$. Then, the performance of $K_S$ and $K_R$ on the real test dataset $D$ is evaluated: a high-fidelity synthetic dataset $S$ will allow $K_S$ to exhibit similar performance to $K_R$ despite never encountering real datapoints before test-time. We report both $K_R$ and $K_S$ in our results, considering MLE to be the difference in performance between $K_R$ and $K_S$.

We use a random forest classifier or regressor as our downstream model $K$. For classification datasets, we compare the accuracy of $K_R$ and $K_S$, while for regression datasets, we compare the coefficient of determination $R^2$. We define the coefficient of determination $R^2$ as $\max(1 - \frac{r}{t}, 0)$, where $r$ and $u$ are the residual sum of squares and total sum of squares, respectively. This formulation means that if a model performs worse than random guessing, its $R^2$ value will be represented as 0. For both the accuracy and $R^2$ coefficient metrics, a higher score indicates higher-quality data. In Appendix B, we also report the mean squared error of $K_R$ and $K_S$ for the regression datasets, where lower scores indicate better performance.

*Detection accuracy* (Qian et al., 2023) quantifies the degree to which the generative model introduces spurious correlations or other patterns that differentiate synthetic from real data. Given the real training dataset $R$ and a synthetic dataset $S$, we sample the same number of rows from each. Next, we train a random forest classifier $C$ to discriminate between real and synthetic examples. Highest-quality synthetic data will result in 50% discrimination accuracy, indicating that $C$ is unable to distinguish between $R$ and $S$. For this reason, our reported discrimination scores are calculated as the absolute difference between 50% and the accuracy of discriminator $C$. Accordingly, lower discrimination scores represent better performance.

**Calculation of results:** The reported result for each model and training setup is the average across three training runs. For each of the three trained models, we sample $10,000$ datapoints, compute all evaluation metrics separately for the three resulting synthetic datasets, then calculate the average metric value across all runs. For LLM approaches, each model is trained for up to 50 epochs, using early stopping when the validation loss (assessed every 5000 steps) fails to improve twice in a row. We perform grid search to select the learning rate with lowest validation loss for each model and training setup. For non-LLM works, we follow the procedures detailed in each of these works.

### 4.1 TABBY VERSUS BASELINE SYNTHESIS METHODS

We begin by validating our first claim.
**Claim 1**: Tabby models generate higher-quality data than pre-existing tabular synthesis approaches.

**Setup**: The left side of Table 2 summarizes the MLE results for each dataset, with the classification datasets (Diabetes, Travel and Adult) on the top and the regression datasets (Abalone, Rainfall and House) on the bottom. For the classification datasets, the reported metric is the accuracy of the downstream random forest classifier, while for the regression datasets, we report the coefficient of determination $R^2$ of the downstream random forest regressor.

For each dataset, the "Original" row corresponds to the "real" MLE achieved by training the downstream classifier or regressor $K_R$ on the original training data $R$ instead of synthetic data. We consider this row to be a performance ceiling for synthetic approaches. Any model and training technique that achieves MLE equal to or better than the "Original" row is considered to be a top-performing approach and is presented in boldface.

**Results**: We find that ***Tabby models achieve the highest MLE in 4 out of 6 datasets***. Further, Tabby reaches upper-bound performance on Diabetes, Travel and Adult, indicating that *Tabby synthetic data is a capable stand-in for real data* in scenarios similar to the MLE task for these datasets.

Lower-performing LLM architectures often experience a performance boost over Plain training when trained using GReaT (either alone or with TapTap and Tabula). However, we find that the increased modeling capacity of Tabby MH allows this model to achieve the best LLM performance

Table 2: Machine Learning Efficacy (MLE). The reported metric is random forest downstream accuracy for classification datasets, or the $R^2$ coefficient for regression datasets. The "Original" row is upper-bound performance. Top results (or any result higher than upper-bound) is presented in bold. An asterisk indicates that at least one of three runs did not successfully produce valid samples. Tabby model names are presented in italic. Highlighted rows correspond to the overall top-performing prior work (Tab-DDPM) , LLM-based work (GTT NT) , and *Tabby model (Plain MH) (Ours)* . Tabby achieves strong performance, reaching upper-bound performance on $4/6$ datasets.

| | | **MLE** (↑) | | | **Discrimination** (↓) | | |
| | | Diabetes | Travel | Adult | Diabetes | Travel | Adult |
|---|---|---|---|---|---|---|---|
| | Original | 73.3 | 87.5 | 84.5 | | | |
| | CTGAN | $52.1 \pm 12.0$ | $61.9 \pm 33.0$ | $76.2 \pm 0.0$ | $40.0 \pm 2.2$ | $32.2 \pm 5.7$ | $47.9 \pm 0.3$ |
| | TVAE | $62.2 \pm 0.0$ | $63.9 \pm 29.6$ | $80.5 \pm 0.6$ | $45.3 \pm 1.6$ | $49.5 \pm 0.2$ | $46.7 \pm 1.7$ |
| | Tab-DDPM | $71.9 \pm 4.5$ | $\mathbf{88.9 \pm 1.4}$ | $83.9 \pm 0.3$ | $11.5 \pm 0.3$ | $\mathbf{1.4 \pm 1.0}$ | $\mathbf{0.9 \pm 0.4}$ |
| Plain | NT | $\mathbf{75.3 \pm 1.5}$ | $85.5 \pm 1.7$ | $\mathbf{84.5 \pm 0.4}$ | $\mathbf{3.8 \pm 1.2}$ | $2.6 \pm 2.0$ | $8.2 \pm 1.0$ |
| Plain | *MMLP* | $\mathbf{74.8 \pm 3.4}$ | $83.7 \pm 2.5$ | $77.4 \pm 1.4$ | $20.6 \pm 2.1$ | $3.4 \pm 2.1$ | $33.6 \pm 8.9$ |
| Plain | *MH* | $\mathbf{74.3 \pm 0.4}$ | $\mathbf{87.7 \pm 1.2}$ | $\mathbf{84.5 \pm 0.2}$ | $6.0 \pm 2.1$ | $3.0 \pm 2.5$ | $9.8 \pm 0.8$ |
| Plain | *MMLP-MH* | $68.1 \pm 0.7$ | $82.5 \pm 1.5$ | $76.6 \pm 0.6$ | $18.6 \pm 2.0$ | $2.2 \pm 1.0$ | $31.4 \pm 11.6$ |
| GReaT | NT | $62.2 \pm 0.7$ | $87.3 \pm 0.9$ | $82.9 \pm 1.1$ | $27.8 \pm 0.7$ | $5.6 \pm 1.7$ | $20.4 \pm 1.0$ |
| GReaT | *MMLP* | $\mathbf{73.8 \pm 0.9}$ | $86.9 \pm 0.6$ | $83.2 \pm 0.6$ | $22.7 \pm 1.2$ | $6.1 \pm 1.5$ | $18.4 \pm 0.9$ |
| GReaT | *MH* | $63.7 \pm 1.3$ | $86.3 \pm 2.1$ | $83.2 \pm 0.1$ | $29.5 \pm 2.1$ | $6.5 \pm 0.4$ | $20.1 \pm 1.7$ |
| GReaT | *MMLP-MH* | $69.4 \pm 3.7$ | $86.7 \pm 0.3$ | $83.0 \pm 0.2$ | $24.3 \pm 1.3$ | $7.0 \pm 3.4$ | $19.6 \pm 0.8$ |
| GTT | NT | $71.9 \pm 5.9$ | $87.1 \pm 1.5$ | $82.9 \pm 0.8$ | $27.1 \pm 1.6$ | $5.6 \pm 0.7$ | $20.5 \pm 1.7$ |
| GTT | *MMLP* | $69.4 \pm 4.3$ | $86.9 \pm 0.6$ | $83.4 \pm 0.3$ | $27.7 \pm 1.1$ | $8.0 \pm 1.0$ | $17.8 \pm 0.8$ |
| GTT | *MH* | $62.5 \pm 0.4$ | $85.7 \pm 0.6$ | $77.1 \pm 5.3$ | $28.1 \pm 1.6$ | $5.2 \pm 0.8$ | $26.2 \pm 6.6$ |
| GTT | *MMLP-MH* | $\mathbf{75.3 \pm 1.5}$ | $85.7 \pm 1.2$ | $83.0 \pm 0.5$ | $24.2 \pm 1.4$ | $7.1 \pm 2.8$ | $18.3 \pm 0.3$ |

| | | **MLE** (↑) | | | **Discrimination** (↓) | | |
| | | Abalone | Rainfall | House | Abalone | Rainfall | House |
|---|---|---|---|---|---|---|---|
| | Original | 0.53 | 0.70 | 0.81 | | | |
| | CTGAN | $0.00 \pm 0.00$ | $0.00 \pm 0.00$ | $0.00 \pm 0.00$ | $46.0 \pm 0.3$ | $18.6 \pm 5.4$ | $18.2 \pm 5.9$ |
| | TVAE | $0.03 \pm 0.05$ | $0.00 \pm 0.00$ | $0.08 \pm 0.13$ | $45.1 \pm 1.6$ | $40.0 \pm 1.8$ | $10.9 \pm 2.6$ |
| | Tab-DDPM | $\mathbf{0.52 \pm 0.01}$ | $\mathbf{0.60 \pm 0.01}$ | $0.59 \pm 0.00$ | $\mathbf{2.7 \pm 0.3}$ | $\mathbf{1.1 \pm 0.9}$ | $33.2 \pm 3.8$ |
| Plain | NT | $0.46 \pm 0.01$ | $0.41 \pm 0.35$ | $0.70 \pm 0.11$ | $5.3 \pm 0.8$ | $3.1 \pm 0.5$ | $6.7 \pm 5.7$ |
| Plain | *MMLP* | $0.28 \pm 0.10$ | $0.11 \pm 0.10$ | $0.00 \pm 0.00$ | $19.9 \pm 2.9$ | $11.8 \pm 1.0$ | $20.2 \pm 4.4$ |
| Plain | *MH* | $0.47 \pm 0.01$ | $0.58 \pm 0.03$ | $\mathbf{0.75 \pm 0.00}$ | $6.2 \pm 0.3$ | $7.1 \pm 1.2$ | $\mathbf{3.8 \pm 0.6}$ |
| Plain | *MMLP-MH* | $0.30 \pm 0.04$ | $0.28 \pm 0.24$ | $0.00 \pm 0.00$ | $21.3 \pm 2.7$ | $11.3 \pm 2.0$ | $23.8 \pm 1.8$ |
| GReaT | NT | $0.39 \pm 0.01$ | N/A* | $0.67 \pm 0.02$ | $8.5 \pm 2.0$ | N/A* | $18.4 \pm 4.2$ |
| GReaT | *MMLP* | $0.34 \pm 0.02$ | $0.16 \pm 0.28$ | $0.68 \pm 0.01$ | $7.5 \pm 1.2$ | $27.2 \pm 16.5$ | $16.1 \pm 0.2$ |
| GReaT | *MH* | $0.36 \pm 0.04$ | $0.00^*$ | $0.67 \pm 0.01$ | $10.7 \pm 3.1$ | $35.7^*$ | $19.1 \pm 0.8$ |
| GReaT | *MMLP-MH* | $0.34 \pm 0.04$ | $0.16 \pm 0.28$ | $0.68 \pm 0.01$ | $7.0 \pm 0.5$ | $25.3 \pm 18.8$ | $16.3 \pm 0.8$ |
| GTT | NT | $0.35 \pm 0.02$ | $0.00 \pm 0.00$ | $0.68 \pm 0.02$ | $\mathbf{5.3 \pm 1.0}$ | $30.4 \pm 9.9$ | $18.6 \pm 2.0$ |
| GTT | *MMLP* | $0.36 \pm 0.01$ | $0.00^*$ | $0.67 \pm 0.01$ | $14.0 \pm 1.8$ | $37.2^*$ | $19.0 \pm 0.9$ |
| GTT | *MH* | $0.36 \pm 0.01$ | $0.26 \pm 0.37$ | $0.66 \pm 0.00$ | $16.2 \pm 0.9$ | $19.4 \pm 15.0$ | $19.2 \pm 0.7$ |
| GTT | *MMLP-MH* | $0.40 \pm 0.04$ | $0.08 \pm 0.11$ | $0.68 \pm 0.01$ | $14.2 \pm 2.0$ | $22.2 \pm 6.6$ | $15.7 \pm 1.0$ |

without the use of additional training techniques, with their associated implementation difficulty and computational overheads. *Plain-trained Tabby MH models demonstrate the highest MLE among all LLM architectures and training styles*.

We also find that pre-existing LLM tabular training techinques introduce undesirable effects on the Rainfall dataset. Entries marked by an asterisk (*) for this dataset indicate that at least one of the three runs for this setup was unsuccessful in synthesizing *any* valid samples. Particularly, the Non-Tabby (NT) GReaT model was unable to produce valid samples in any of the runs. Meanwhile, each Plain-trained model is successfully sampled in all three runs—and the Plain-trained Tabby MH models achieves the best overall performance on this dataset, indicating that **Tabby models**

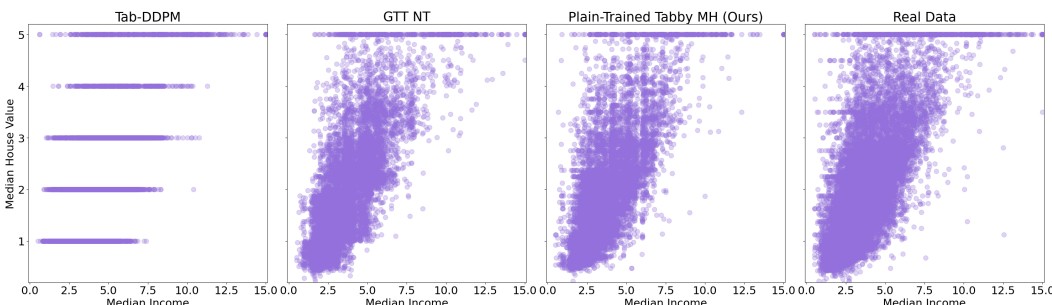

Figure 2: The House dataset's target Median House Value column as a function of its most-predictive feature, Median Income. Left to right: synthetic data from Tab-DDPM, the prior best LLM-based method and Plain Tabby MH, followed by the original data distribution.

**are capable of modeling complexities within the Rainfall dataset that pre-existing LLM-based tabular synthesis works are unable to capture**.

We view similarly-desirable performance in Tabby through the discrimination metric on the right side of Table 2. Lower scores indicate that the discrimination classifier is less able to distinguish between real and synthetic data samples. Comparing the discrimination scores of Plain- and GReaT-trained NT, we find that the application of specialized tabular training techniques often has the unfortunate side-effect of worsening the discrimination score. For example, the discriminator score on Diabetes for NT is $24\%$ worse under GReaT than with Plain. Meanwhile, we see that the discrimination scores for Plain-trained models do not increase as much. The discriminator score for Diabetes increases only $2.2\%$ for Plain-trained Tabby MH over the NT model.

### 4.1.1 COMPARING MULTIVARIATE MODELING CAPABILITIES ACROSS MODELS

We further compare the multivariate modeling capabilities of Tab-DDPM, Plain-trained Tabby MH and the prior top-performing LLM-based approach of Great-trained NT with TapTap and Tabula. For the real datapoints, plus synthetic data from one run of each model on the House dataset, Figure 2 plots the target column (Median House Value) as a function of its most predictive feature in the dataset (Median Income).

Tab-DDPM's plot (leftmost) differs the most from the real data (rightmost) because this model only supports integer-valued regression targets. As a result, both LLM-based approaches more accurately capture the target column's distribution than Tab-DDPM.

Meanwhile, GReaT sampling (center left) enforces that the target column's distribution is replicated in synthetic data by prompting the model with target values selected randomly, based on their frequency in the training data. This constraint means that GReaT models will not generate target values outside those in the training data, which can be undesirable for datasets with few rows or limited target column coverage. In contrast, Plain training (center right) allows the language model to generate previously unseen target values, while the improved modeling capacity of the Tabby over the NT model allows this approach to still effectively capture the overall distribution of the target column.

### 4.2 INVESTIGATING THE CHOICE OF BASE MODEL

We now turn to our second claim.
**Claim 2**: The Tabby architecture modification allows smaller LLMs to achieve synthetic data fidelity more similar to that of LLMs with higher parameter counts.

**Comparisons:** Next, we compare Non-Tabby and Tabby MH performance for LMs of varying sizes. As a base model, we use either Distilled-GPT2 with $80$ million parameters or Llama 3 with $8$ billion parameters. Each model is trained using GReaT for 5 epochs on a subset of the House dataset with six columns and 5160 rows. The Llama models are fine-tuned and sampled using LoRA on all linear linear layers, including the LM head to accommodate the new tokens added prior to fine-tuning and—for the Tabby model—the MH modification.

Table 3: A comparison of MLE and discriminator scores across models of varying size. We observe that while Tabby increases model performance in both large and small models, the performance increase is more noticeable for Distilled-GPT2. Additionally, we see that synthetic data generated by Llama models is more easily distinguished than Distill GPT-2 synthetic data from real data.

| | MLE | Discrimination | # Parameters |
|---|---|---|---|
| Real | 0.662 | | |
| NT DGPT-2 | $0.474 \pm 0.022$ | $16.4 \pm 0.5$ | 80 Million |
| *MH DGPT-2* | $0.525 \pm 0.028$ | $\mathbf{16.3 \pm 0.7}$ | 270 Million |
| NT Llama | $0.560 \pm 0.015$ | $24.2 \pm 1.6$ | 8 Billion |
| *MH Llama* | $\mathbf{0.562 \pm 0.022}$ | $25.3 \pm 1.2$ | 10.5 Billion |

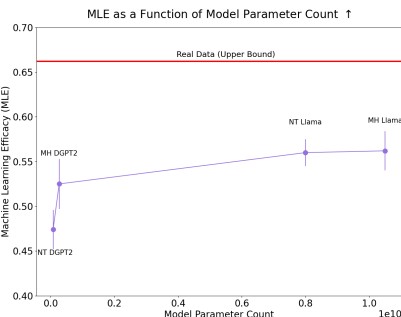 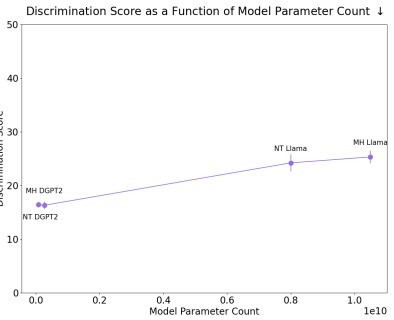

Figure 3: Machine Learning Efficacy (MLE) and Discrimination Score as a function of parameter count. MH Tabby DGPT2 splits the performance difference between NT DGPT2 and Llama, despite having a similar parameter count (270M) to NT DGPT2 (80M).

**Results:** As depicted in Table 3 and plotted in Figure 3, both Llama-based models outperform the Distilled-GPT2-based models on MLE. Additionally, the Tabby MH models outperform their Non-Tabby counterparts on MLE, though the difference is more pronounced for the Distilled-GPT2-based models. In fact, with an average MLE of $0.525$, **the 270 million-parameter Tabby MH Distilled-GPT2 model achieves closer performance to the 8 billion-parameter Non-Tabby Llama model than to the Non-Tabby Distilled-GPT2 model with 80 million parameters**.

We further observe an interesting effect when comparing MLE and discrimination scores. While Llama achieves higher MLE, the Distilled-GPT2-based models both achieve better discrimination scores. The double-edged nature of expressivity is one possible explanation for this phenomenon: While higher expressivity allows Llama-based models to fit to complex patterns occurring in the real data, perhaps this expressivity also allows Llama-based models to inject additional, spurious patterns which differentiate the synthetic from the real data—resulting in a worse discrimination score than the smaller, Distilled-GPT2-based models.

### 4.3 TRACKING THE ADAPTATION TO INDIVIDUAL COLUMNS

We address our final claim by examining Tabby's progress while fine-tuning on tabular data.

**Claim 3**: Tabby's loss formulation allows for convenient tracking of per-column performance at training time, leading to better understanding of model behavior.

**Setup:** For three runs, we train a Tabby MH model on a subset of the House dataset containing 5160 rows and six columns. We log the individual columns' losses on the evaluation dataset every 2500 steps while training for 10 epochs, then average across the runs and display the results in Figure 4.

**Results**: We observe that Occupancy is the largest contributor to the model's loss until step 32000. While Median Income's loss is initially the second-lowest, it improves little throughout the training process and exhibits the highest loss of all columns at the end of training. Additionally, we view that convergence occurs around step 40000.

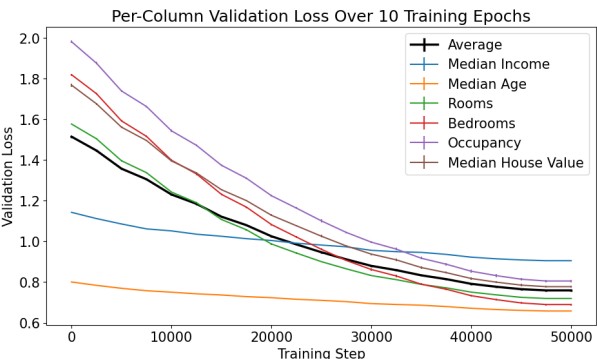

Figure 4: Per-column validation loss across 10 epochs of training Tabby MH Distilled-GPT2 on a subset of House, with average validation loss (black line). While the Occupancy column initially displays the highest loss, Median Income improves little throughout training and becomes the highest-loss column by step 32000.

These insights are useful in cases where the model struggles to learn some columns more than others. Such information may indicate a need for better preprocessing for a difficult column, or gathering more datapoints to demonstrate the column's distribution. Additionally, the ability to track each column's loss individually and to determine that the losses are roughly balanced across columns, rather than very low in some columns and very high in others, may improve trust in the model—we can understand that there is a low, aleatoric error in each column as opposed to a sizeable epistemic error in a few columns.

### 4.4 DISCUSSION

We find that Tabby models synthesize high-quality data in a variety of settings. In particular, Plain-trained Tabby MH consistently outperforms all prior LLM-based approaches. The Tabby architecture modification allows LLMs to better model univariate column distributions, as well as the multivariate relationships across columns. This impact is particularly evident in smaller models, as demonstrated in Section 4.2. Additionally, Tabby's loss formulation allows per-column performance to be evaluated more easily than other deep learning-based tabular approaches, which may be useful in selecting training techniques and strengthening trust in model performance.

Unusually, we find that the baseline Plain training technique with Distilled-GPT2 achieves near-optimal MLE performance on several datasets. The exclusion of the Plain training technique from prior LLM works for tabular data synthesis, which rely on more complex techniques, is surprising.

As of this writing, the Adult, House and Diabetes datasets have become quite prevalent evaluation tasks for tabular synthesis works: these three datasets are common to most works in this area. It is our recommendation that new standard evaluation datasets be identified, which pose greater challenges for baselines than these three datasets. A focus on more challenging datasets will allow researchers to better-demonstrate the value of novel and more-complex synthesis techniques.

## 5 CONCLUSION

We introduce Tabby, a Mixture-of-Experts-based architecture modification that allows LLMs to be better suited to tabular datasets. Tabby reaches parity with non-synthetic data in two out of three evaluated datasets, according to machine learning efficacy with a Decision Tree Classifier. Given the promising performance of Tabby, we hope to spur future work in this area and further experimentation with architecture modifications that allow LLMs to better fit to tabular data. The concepts behind Tabby may find utility in similar modalities as well, such as geospatial, nested-list, or other highly-structured data.

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
