# OpenReview forum: "Tabby: Tabular Adaptation for Language Models"
_ICLR.cc/2025/Conference — Submitted to ICLR 2025_

### Official Review · Reviewer_RYWz · 2024-10-31

**Soundness:** 2
**Presentation:** 2
**Contribution:** 2
**Rating:** 5
**Confidence:** 3

**Summary:**

In this paper, the authors present a tabular data synthesis approach, Tabby. The novelty of Tabby lies in two main aspects: (1) modifying the original transformer model by applying MoE-like techniques to better model tabular data, and (2) designing a specialized data format for tabular data. Experimental results show that Tabby achieves comparable performance to the previous state-of-the-art, Tab-DDPM, and outperforms GTT NT.

**Strengths:**

- The model modifications and data organization are well-motivated and intuitive.
- The distribution of the synthesized data is very close to the natural data.
- The experimental results looks good.

**Weaknesses:**

- Tabby seems achieve comparable results to Tab-DDPM with marginal performance gain in Table 2.
- The method is quite simple and not much effective in final performance.

**Questions:**

Q1: Have you computed the FLOPs for training on different datasets? It seems that Tabby uses a fixed pattern to organize tabular data, which may require more tokens for computation.

Q2: Regarding Claim 2, could you provide a scaling curve showing performance relative to model size or data quantity? It would be interesting to see how Tabby impacts different models and how the amount of Tabby data influences the learning process. Additionally, a comparison of the scaling curve between Tabby data and natural data would serve as evidence of Tabby data being a scalable alternative to natural data.

Q3: I'm not sure if the modification to original network is necessary. Is there an ablation study?

---

> ### Author Response · Authors · 2024-11-20
>
> We appreciate your feedback and careful attention to our paper! We are very happy with the promising results demonstrated by Tabby and grateful that you have noted this effect in your review. To address your comments:
> 1. **On performance gain**. We note that **_Tabby achieves similar results to Tab-DDPM, but with far fewer assumptions on the structure of the dataset_**. For example, as demonstrated in Figure 2 and discussed in Section 4.1.1, Tab-DDPM is not capable of generating real-valued numbers in the label column. In other words, Tab-DDPM is only capable of synthesizing regression datasets with integer-valued labels, which is a significant limitation that Tabby fully circumvents through its LLM architecture design. Additionally, Tab-DDPM is limited in its support for diverse column modalities. For instance, we have applied Tabby to a private dataset that contains a column of IP addresses (i.e. strings of the format 01.234.567.89 and similar). Tabby and other LLM-based tabular synthesis architectures are fundamentally capable of learning to synthesize these formatted strings and create novel IP addresses not occurring in the training dataset, whereas Tab-DDPM must model the IP column as categorical and can only synthesize datapoints with IP addresses that occur in the training data. This limitation poses severe privacy implications for datasets that contain columns such as home addresses, hospital patient names, telephone numbers and more. Tabby achieves similar performance to Tab-DDPM, while removing these limitations on column modalities.
> 2. **On simplicity**. While simple, Tabby is **_the first LLM architecture modification for tabular data synthesis_**. While the MoE approach is intuitive and achieves good results, as demonstrated in our evaluations and considering the modality limitations of the similarly-performing prior work (Tab-DDPM), the implementation of architecture modifications in large pretrained models is often far from simple. As we share with Reviewer MewM, the Tabby architecture modification will serve as the inspiration and starting point for additional architectural modifications to support modalities other than tabular data, including relational and geospatial data. We believe we have just scratched the surface here, and that experimenting with further modifications to support additional structured data modalities will produce additional value.
>
> As for your questions:
> - **On computation**. Tabby **_does not_** require any more tokens than the prior LLM works in our main results. In fact, our way of organizing tabular data is drawn from GReaT, and is also used in TapTap and Tabula.
> - **On scaling**. Our Section 4.2, “Tabby Performance as a Function of Base Model Size”, addresses how Tabby impacts different models (Distilled-GPT2 versus 8 billion parameter Llama 3). It finds that Tabby Distilled-GPT2's MLE performance is over halfway between the MLE performances of non-Tabby Distilled-GPT2 and Llama ($\frac{\texttt{MLE(Tabby DGPT2)}-\texttt{MLE(Non-Tabby DGPT2)}}{\texttt{MLE(Non-Tabby Llama)}-\texttt{MLE(Non-Tabby DGPT2)}} \approx 59.30\\%$), while Tabby Distilled-GPT2's increase in parameter count compared to non-Tabby Distilled-GPT2 is only about 2% of 8B Llama's increase in parameter count compared to non-Tabby Distilled-GPT2 ($\frac{\texttt{params(Tabby DGPT2)}-\texttt{params(Non-Tabby DGPT2)}}{\texttt{params(Non-Tabby Llama)}-\texttt{params(Non-Tabby DGPT2)}} \approx 2.39\\%$). We have updated our paper to include a plot of this scaling curve (Figure 3 in main text). Meanwhile, the Machine Learning Efficacy (MLE) metric in our main results (defined in Section 4.0.3 with results in Section 4.1 and Table 2) compares the performance of a downstream model trained fully on real versus synthetic data for each of the evaluated synthesis approaches. According to this metric, Tabby synthetic data achieves equivalent performance to real data in 3/6 evaluated datasets. While MLE is the standard metric by which table synthesis methods are evaluated, the exploration of a metric including a downstream model trained both on some real and some synthetic data could be an interesting area of future work.
> - **On ablations**. Our main results table (Table 2) and analysis in Section 4.1 compares the performance of a plain-trained non-Tabby LLM and a plain-trained Tabby LLM. We are happy to provide any additional ablations.
>
> We thank you again for your time and thoughts!

---

> > ### Author Response · Authors · 2024-11-26
> >
> > Dear Reviewer,
> >
> > We thank you again for your feedback, questions, and suggestions! We believe we have answered all of your questions in our response. If you have additional questions, we would love to answer them!
> >
> > Sincerely,
> > The Authors

---

### Official Review · Reviewer_MewM · 2024-11-02

**Soundness:** 3
**Presentation:** 3
**Contribution:** 2
**Rating:** 3
**Confidence:** 5

**Summary:**

This work introduces a new model called Tabby for tabular data. Tabby is an architecture modification that enables transformer-based language models to synthesize more realistic tabular data. It introduces Gated Mixture-of-Experts layers to better model the complex interdependencies and diverse data types found in tabular datasets. Tabby outperforms previous tabular data synthesis methods, achieving outstanding performance on multiple benchmarks.

**Strengths:**

1. Tabby achieves strong performance in benchmark evaluation. It generates high-quality synthethic tabular data in comparison with the baseline methods.
2. The introduction of MoE shows effectiveness in helping the model understand tabular data structure and generate higher-quality tabular data.

**Weaknesses:**

1. The design of MoE layer is complex. For a table with V columns, this article should design an MoE model with V experts to adapt to the table. This is not generalizable to data of diverse formats. It is suggested to modify the model design to be more compatible and more generalizable.
2. Scalable experiments are advised to be conducted. This study needs to provide experimental results on datasets of larger scales and also more commonly used datasets.
3. The experiments are advised to be conducted on contemporary large language models, including Llama, Qwen, Mistral, instead of Distilled-GPT2.

**Questions:**

1. Have you conducted experiments on the recently released large language models? If yes, which model sizes did you choose?

---

> ### Author Response · Authors · 2024-11-20
>
> Thank you for taking the time to read our paper and sharing your thoughts! We are very excited to share our findings on the usefulness of MoE for the tabular synthesis community. To address your comments:
> 1. **On MoE design**. Indeed, when we have $V$-column data, we make a model with $V$ experts. This design is generalizable to any tabular dataset, where these are defined as any dataset consisting of $V$ columns and $N$ rows. While all prior related works to Tabby focus uniquely on tabular data as well, we note that Tabby **_can easily be generalized_** to be compatible with even more modalities that are similar to tabular data. For example, hierarchical or tree-structured data can be addressed with the same underlying design— the main challenge is not specifically the structure, but rather the potential complexity from adding very large numbers of heads. However, this can be circumvented by using parameter sharing. For example, in the hierarchical case such as JSON objects, we can create nested MoE layers: the outer-level objects are modeled by the outer-level experts, which each are blocks comprised of inner-level experts that model the inner-level objects.
> 2. **On scalability**. Our choice of datasets for our experiments is **_simply based on the standard for tabular synthesis works, enabling for easy comparison_**. In particular, most works in our main results table hinge their analyses on the Adult, House and Diabetes datasets, which we center in our paper. Here are the datasets used in each of the prior publications that we compare against in our main results:
>     - [GReaT](https://arxiv.org/abs/2210.06280): *Adult, House, Diabetes*, Travel, Sick, HELOC
>     - [TapTap](https://arxiv.org/abs/2305.09696): *Adult, House, Diabetes*, HELOC, Credit score, Loan, Dubai housing, Crab age, Medical cost, Gem, Beam, Sick
>     - [Tabula](https://arxiv.org/abs/2310.12746): *Adult*, Loan, Covtype, Intrusion, King, Insurance
>     - [Tab-DDPM](https://arxiv.org/abs/2209.15421): *Adult, House, Diabetes*, Abalone, Buddy, Cardiovascular, Travel, Facebook, Gesture, Higgs, House 16H, Insurance, King, MiniBooNE, Wilt
>     - [CTGAN/TVAE](https://arxiv.org/abs/1907.00503): *Adult, House*, Covertype, Intrusion, News
> - In our draft’s Section 4.4 (around line 515), we shared similar thoughts as you mention in your review: that the standard tabular synthesis benchmark datasets should include more challenging data. In our paper, **_we worked to include higher-diversity evaluation datasets than the prior works by including additional regression datasets (Abalone and Rain)_** while still allowing our evaluation setup to be comparable to prior works. Regression datasets have also been largely overlooked in prior tabular synthesis works, which often use only the House dataset as a representative for all regression tasks. Our main results table demonstrates that the synthesis methods (prior works included) struggle even more on regression datasets than on classification ones. This issue is further demonstrated by our Figure 2 and Section 4.1.1, which demonstrate that **_prior works require notable assumptions to generate regression data, which Tabby models do not_**.
> 3. **On LLM usage**. We agree! **_We conduct experiments using larger models (LLaMA 3 8B)_** and include them in the main section of our paper (Section 4.2). In the main results (Table 2 and Section 4.1), we work with Distilled-GPT2 because the prior works on tabular synthesis with LLMs (GReat, TapTap and Tabula) all implement Distilled-GPT2. As such, this choice allows for maximum comparability with the prior works.
>
> In reference to your Question, please also refer to our paper’s Section 4.2, “Investigating the Choice of Base Model”.
>
> Thank you again for sharing your thoughts on our paper and we would appreciate any additional feedback!

---

> ### Author Response · Authors · 2024-11-26
>
> Dear Reviewer,
>
> We thank you again for your feedback, questions, and suggestions! We believe we have answered all of your questions in our response. If you have additional questions, we would love to answer them!
>
> Sincerely,
> The Authors

---

### Official Review · Reviewer_mxT7 · 2024-11-04

**Soundness:** 1
**Presentation:** 2
**Contribution:** 1
**Rating:** 1
**Confidence:** 3

**Summary:**

The paper proposes to use a MOE LLM fine tuned on table data for synthetic table data synthesis.  The authors find that their method outperforms previous methods on table synthesis benchmarks.

**Strengths:**

The proposed method outperforms other methods on most datasets and metrics presented.

**Weaknesses:**

* There is no significant difference between tabby and the non-tabby (I assume no MOE?) baseline.  Given that MOE has a lot more parameters, this is a negative finding.
* The papers contributions are very minor - applying MOE to a narrow problem (table generation).  And the results are not all that strong.
* It's not easy from the presentation what exactly do the tasks require, what exactly are the baselines and model variations.

**Questions:**

Can you please detail the various architectures MMLP, MH and MMLP+MH?
Why does MMLP+MH underperform, even though it is more complex?
Do you replace every layer with MOE?

---

> ### Author Response · Authors · 2024-11-20
>
> Thank you for your feedback on Tabby. We respond to each of your points below.
> 1. **On performance compared to Non-Tabby models**. Indeed, Non-Tabby means that we do not use MoE (refer to definition in Section 3.1 for details). **_Our main results in Table 2 and Section 4.1 demonstrate that Plain MH Tabby models outperform the prior best LLM-method (GTT Non-Tabby) on each of the six datasets that we evaluate_** in the primary metric of Machine Learning Efficacy (MLE). The Plain MH Tabby model reaches upper-bound performance in 3/6 datasets, whereas the GTT Non-Tabby model does not reach upper-bound in any of the datasets. Furthermore, **_our Plain MH Tabby model outperforms the Plain Non-Tabby model on five out of six datasets_** (both Plain MH Tabby and Plain Non-Tabby reach upper-bound performance on the sixth dataset, Adult) on MLE. According to these results, **_the Plain MH Tabby model significantly outperforms all preexisting LLM-based tabular synthesis methods_**.
> 2. **On significance of contribution**. As stated in Section 1 (Introduction), **_there have been “many calls for improved tabular approaches” [1,2,3]_**. Tabular data is critical for many domains, and the development of high-fidelity tabular synthesis methods will have important implications such as improved data augmentation and missing value imputation for tabular tasks, as well as the ability to generate synthetic datasets that will preserve the privacy of original datasets in domains such as medicine. Furthermore, **_the tabular modality is part of a broader class of structured modalities_**, including data in nested or tree structures. Positive findings in the area of tabular data, such as Tabby, will spur future progress in the broader modalities of structured data as well.
> 3. **On definitions of tasks, baselines and model definitions**. We address each of these in-turn:
>     - **Tasks**: **_We use the standard evaluation tasks defined by prior tabular synthesis works_**, such as [GReaT](https://arxiv.org/abs/2210.06280) and [Tab-DDPM](https://arxiv.org/abs/2209.15421). While these tasks are described in detail in Section 4.0.3, we provide a brief summary here. The primary evaluation of a tabular synthesis method on a given dataset is **_Machine Learning Efficacy (MLE)_**. Denote the dataset’s train and test splits by $R$ and $D$, respectively. We first train our synthesis method on $R$, then use the synthesis method to create a synthetic dataset $S$. To calculate MLE, a downstream random forest classifier or regressor (determined by the dataset’s *label column*, which is specified in Table 1), denoted $K_R$, is trained on $R$ to predict some predetermined label column. Another classifier or regressor, denoted $K_S$ is trained on $S$ to predict the same label column. Then, both $K_R$ and $K_S$ are evaluated on the test dataset $D$. The difference in test-time performance between $K_R$ and $K_S$ is referred to as MLE. Further details are shared in Section 4.0.3, around lines 274-287, and our secondary metric of Discrimination Score is then detailed in lines 288-295. Our datasets are introduced in Section 4.0.2 and Table 1, with further information such as download links and descriptions of their columns in Appendix A.
>     - **Baselines**: Section 4.0.1, entitled “Baselines and Comparisons”, specifies each of our baselines. In particular, we include the prior LLM training techniques for tabular data (GReaT, TapTap and Tabula, and referred to as GTT when used in-concert with each other), the longstanding popular GAN- and VAE-based approaches (CTGAN and TVAE), and the prior state of the art Tab-DDPM, which uses a diffusion model architecture.
>     - **Model Definitions**: We define each of our architectures in Section 3.1, “Architecture of Tabby Models”, around lines 154-157, and Figure 1 offers a visual comparison of Non-Tabby, Tabby Multi-MLP (MMLP) and Multi-Head (MH) models. For dataset with $V$ columns, compared to a standard, Non-Tabby transformer-based LLM, the Tabby MMLP model replaces the MLP within each of its transformer blocks with an MoE layer of $V$ experts and the Tabby MH model replaces its language modeling head with an MoE layer of $V$ experts. The Tabby MMLP-MH model replaces each of the transformer MLPs and the language modeling head with $V$-expert MoE layers.
>
> (continued in next comment)

---

> ### Author Response · Authors · 2024-11-20
>
> For your questions, please refer to point #3 for each of the model definitions. In our paper, each transformer block MLP is replaced with an MoE layer for the Tabby MMLP and MMLP-MH models, but it is also possible to experiment with only replacing the MLPs in a subset of the transformer blocks. Doing so may offer similar performance to Tabby MMLP or MMLP-MH, but with reduced parameter count.
>
> As for why Tabby MMLP-MH does not perform as well as Tabby MH models, we hypothesize that it may be due to overfitting and memorization of training samples. We have conducted additional experiments since the deadline to determine the extent to which each of the tabular synthesis methods output memorized training samples during synthesis, and **find that Tabby MH models do not memorize at a rate significantly different from the prior works**.
>
> We thank you again for your time and opinions!
>
> 1. B. van Breugel and M. van der Schaar. Why Tabular Foundation Models Should Be A Research Priority, May 2024. URL [https://arxiv.org/abs/2405.01147v2](https://arxiv.org/abs/2405.01147v2).
> 2. M. F. Davila, S. Groen, F. Panse, and W. Wingerath. Navigating Tabular Data Synthesis Research: Understanding User Needs and Tool Capabilities, May 2024. URL [http://arxiv.org/ abs/2405.20959](http://arxiv.org/abs/2405.20959).
> 3. X. Fang, W. Xu, F. A. Tan, J. Zhang, Z. Hu, Y. Qi, S. Nickleach, D. Socolinsky, S. Sengamedu, and C. Faloutsos. Large Language Models (LLMs) on Tabular Data: Prediction, Generation, and Understanding – A Survey, Feb. 2024. URL [http://arxiv.org/abs/2402.17944](http://arxiv.org/abs/2402.17944).

---

> ### Author Response · Authors · 2024-11-26
>
> Dear Reviewer,
>
> We thank you again for your feedback, questions, and suggestions! We believe we have answered all of your questions in our response. If you have additional questions, we would love to answer them!
>
> Sincerely,
> The Authors

---

### Author Response · Authors · 2024-11-20

We thank the attentive reviewers for their feedback and comments on our paper. We have received much valuable feedback that allows us to better explain the advantages of Tabby as compared to prior works, such as its freedom from prior assumptions on data column modalities. We reiterate the contributions and strengths of Tabby here:

- **Tabby achieves state of the art performance on the standard tabular evaluation metrics, with fewer limitations than the prior state of the art method** (Tab-DDPM). Tab-DDPM is unable to generate realistic datasets with real-valued numerical labels and is unable to synthesize strings to the same degree as LLMs, resulting in limitations to the privacy of Tab-DDPM’s synthetic data and the ability of Tab-DDPM to model areas of the data distribution that do not occur in its training data (for further details on these limitations, refer to our comment to Reviewer RYWz). Tabby wholly circumvents these limitations, while achieving comparable state of the art performance and even outperforming Tab-DDPM on some datasets.
- **Tabby is the first architecture modification that allows LLMs to directly generate tabular data**. Because the prior LLM tabular synthesis approaches (e.g. GReaT, TapTap and Tabula) are training techniques, they can be used in concert with Tabby. Further, Tabby is applicable to any transformer-based LLM, as we demonstrate in Section 5.2.
- **Tabby’s Mixture of Experts (MoE) layer design is flexible and may serve as a basis for future work in other structured modalities**, such as JSON objects and geospatial data. While our paper demonstrates the benefits of replacing language modeling heads and transformer block MLPs with MoE layers, there are many other possible combinations of MoE layers. For instance, future work can replace attention heads with MoEs or nest MoEs within each other to generate nested data structures such as JSONs. Tabby’s high performance for tabular synthesis demonstrates that these architecture variations bring large improvements in our ability to generate many varieties of structured data using pre-existing, pretrained architectures such as LLMs.

We again thank our reviewers for the careful attention that they have devoted to our paper, as well as the PC, AC and others who assist in the conducting of ICLR’s rigorous reviewing process.

---

### Meta-Review · Area_Chair_rP92 · 2024-12-19

**Metareview:**

Tabby is a method for transformer-based Large Language Models (LLMs) to synthesize high-fidelity tabular data. Tabby produces higher-quality synthetic data for 4 out of 6 datasets compared to previous methods. An architecture modification is proposed for the task. The reviewers argue that the additional complexity of MoE design is undesirable, and the paper needs more scalable experiments to show generalizability. The paper needs further improvements to make it more solid.

**Additional Comments On Reviewer Discussion:**

The decisions are consistent before and after the rebuttal period. The paper needs a major revision to meet the reviewers' expectation.

---

### Decision · Program_Chairs · 2025-01-22

Reject